# Black-Box Prompt Learning for Pre-trained Language Models

**Shizhe Diao**                                                     *sdiaoaa@connect.ust.hk*
*The Hong Kong University of Science and Technology*

**Zhichao Huang**                                                   *zhuangbx@connect.ust.hk*
*The Hong Kong University of Science and Technology*

**Ruijia Xu**                                                       *rxuaq@connect.ust.hk*
*The Hong Kong University of Science and Technology*

**Xuechun Li**                                                      *xul021@ucsd.edu*
*University of California, San Diego*

**Yong Lin**                                                        *ylindf@connect.ust.hk*
*The Hong Kong University of Science and Technology*

**Xiao Zhou**                                                       *xzhoubi@connect.ust.hk*
*The Hong Kong University of Science and Technology*

**Tong Zhang**[*]                                                   *tongzhang@ust.hk*
*The Hong Kong University of Science and Technology*

**Reviewed on OpenReview:** *https://openreview.net/forum?id=IvsGP7xRvm*

## Abstract

The increasing scale of general-purpose Pre-trained Language Models (**PLMs**) necessitates the study of more efficient adaptation across different downstream tasks. In this paper, we establish a Black-box Discrete Prompt Learning (**BDPL**) to resonate with pragmatic interactions between the cloud infrastructure and edge devices. Particularly, instead of fine-tuning the model in the cloud, we adapt PLMs by prompt learning, which efficiently optimizes only a few parameters of the discrete prompts. Moreover, we consider the scenario that we do not have access to the parameters and gradients of the pre-trained models, except for its outputs given inputs. This black-box setting secures the cloud infrastructure from potential attack and misuse to cause a single-point failure, which is preferable to the white-box counterpart by current infrastructures. Under this black-box constraint, we apply a variance-reduced policy gradient algorithm to estimate the gradients of parameters in the categorical distribution of each discrete prompt. In light of our method, the user devices can efficiently tune their tasks by querying the PLMs bounded by a range of API calls. Our experiments on RoBERTa and GPT-3 demonstrate that the proposed algorithm achieves significant improvement on eight benchmarks in a cloud-device collaboration manner. Finally, we conduct in-depth case studies to comprehensively analyze our method in terms of various data sizes, prompt lengths, training budgets, optimization objectives, prompt transferability, and explanations of the learned prompts.[1]

---

[*]Joint with Google research
[1]The code is available at `https://github.com/shizhediao/Black-Box-Prompt-Learning`.

Table 1: Comparison of different tuning methods. **Frozen**: the pre-trained model is frozen and will not be updated. **Black-Box**: there are no access to the parameters and gradients from the pre-trained model. **Discrete**: the prompts are discrete tokens (compared with soft prompts). **Interpretable**: the prompts are readable and interpretable. **Learnable**: the prompts are parametric and learnable with explicit or estimated gradients (compared with manual prompts). **N/A**: not applicable since the corresponding descriptions are for prompt learning.

| Methods | Frozen | Black-Box | Discrete | Interpretable | Learnable |
|---|---|---|---|---|---|
| Vanilla FineTuning | | | N/A | N/A | N/A |
| GPT-3's FineTuning[3] | ✓ | ✓ | N/A | N/A | N/A |
| FeatureProbe (Peters et al., 2019) | ✓ | | | | ✓ |
| ManualPrompt | ✓ | ✓ | ✓ | ✓ | |
| InContextLearning(Brown et al., 2020) | ✓ | ✓ | ✓ | ✓ | |
| PromptTuning (Lester et al., 2021) | ✓ | | | | ✓ |
| P-Tuning v2 (Liu et al., 2021a) | ✓ | | | | ✓ |
| AutoPrompt (Shin et al., 2020) | ✓ | | ✓ | ✓ | ✓ |
| BBT (Sun et al., 2022) | ✓ | ✓ | | | ✓ |
| BDPL (ours) | ✓ | ✓ | ✓ | ✓ | ✓ |

# 1 Introduction

Large Pre-trained Language Models (PLMs) have demonstrated impressive versatility across a wide spectrum of downstream tasks, via either fine-tuning (FT) (Devlin et al., 2019; Liu et al., 2019; Lewis et al., 2020; Zhang et al., 2020; Yang et al., 2020; Diao et al., 2020; Pan et al., 2022) or prompt-based learning (PL) (Gao et al., 2021; Liu et al., 2021b; Schick & Schütze, 2021; Li & Liang, 2021; Liu et al., 2023). Traditionally, these two tuning paradigms are conducted at a white-box setting, where the parameters and gradients are accessible since the model is usually open-sourced and can be duplicated in user devices. Despite the fact that white-box methods have made remarkable progress, however, the increasing scale of PLMs renders this setting implausible. Nowadays, huge PLMs opt to serve users as commercial APIs deployed in the cloud, such as OpenAI GPT-3[2]. In particular, the service providers hide their model parameters and expose the query and prediction interface, which is termed the black-box setting in this paper.

Although solving NLP problems with APIs in a black-box setting is considerably challenging, it is indeed aligned with the new norm of the current interplay between the cloud infrastructure and edge devices. Specifically, from the position of cloud providers, it is reasonable to restrict the access of pre-trained model parameters since commercial, ethical, legal, security, and other concerns might be raised (Bommasani et al., 2021). First, under the white-box setting, the weaknesses and biases rooted in the underlying PLMs are at higher risk of being misused for harmful purposes. Second, the centralizing nature of PLMs exposes them to potential attacks to cause a single-point failure (Krishna et al., 2020). To this end, the black-box setting is more convincing to secure the cloud infrastructure from being condemned. As for the interests of user devices, the black-box paradigm grants them a more economical option. Otherwise, if we have access to the model's gradients, it requires transmitting gradients from cloud to device, causing high transmission costs.

With this basic setting in mind, we further elaborate on a more pragmatic scenario, i.e., the discrete prompt learning under the black-box constraint. Particularly, we opt for the prompt learning mechanism instead of the fine-tuning counterpart, partially due to the fact that the prompt learning is more cost-effective by tuning fewer parameters and can eliminate the gap between pre-training and downstream transfer. Moreover, black-box fine-tuning[3] requires users to upload their private labels and save the fine-tuned model on the server, which strongly relies on the cloud provider as the single-point trust for the security of private data and model. On the contrary, our black-box prompt learning allows users to store the private label and tune the prompt locally, preventing potential data leakage and protecting the users' commercial interests. For instance, in our setting, each user device can query the output from the cloud and then update its prompts separately on its own data.

It is noteworthy that we optimize discrete prompts, which are more interpretable to power users from different backgrounds to develop their own applications with PLMs. On the contrary, continuous prompt learning

---
[2]https://openai.com/api/
[3]https://beta.openai.com/docs/guides/fine-tuning

methods, e.g., BBT (black box tuning (Sun et al., 2022)), are difficult to interpret their learned prompts. Moreover, these methods fail to be directly applied to prediction APIs because APIs only accept discrete inputs (e.g., GPT-3). However, the discrete nature of our BDPL allows commercial prediction APIs to directly take the learned prompt tokens without pain. Overall, our established Black-Box Discrete Prompt Learning (BDPL) is closely in accordance with the recent progress of huge PLMs, whose comparisons with the existing settings can be found in table 1. As shown in the table, BDPL can be specified under the constraints that the PLMs are frozen and both their parameters and gradients are invisible and share the virtue of optimizing discrete, interpretable, and learnable prompt tokens simultaneously.

To embrace this paradigm shift of tuning PLMs, we design a policy gradient inspired framework that can be optimized without relying on the parameters and gradients of the pre-trained models. Specifically, we characterize the prompt learning procedure as a discrete token selection problem, where the proper prompt tokens are sampled according to a categorical distribution. Because the PLM's parameters are invisible and gradients cannot be back-propagated, the categorical distribution needs to be optimized by some gradient-free algorithms. We resort to the policy gradient algorithm to estimate the gradients without back-propagation. Moreover, to eliminate the high variance issue of policy gradient, we adopted a variance-reduced policy gradient estimator.

Experimental results on two kinds of datasets, *i.e.*, datasets without domain-shift and datasets with domain-shift, demonstrate the effectiveness of the proposed black-box discrete prompt learning, which significantly improves the performance over a generic pre-trained model and outperforms all baseline models on eleven datasets. The results confirm that incorporating black-box prompt learning for pre-trained models is an effective and efficient solution to the PLM adaptation. We also present further analyses by investigating the effects of different training data sizes, prompt lengths, training budgets, and objectives. Our analyses demonstrated the robustness, scalability, and transferability of the proposed method.

The contributions of our work are as follows:

• We propose a new setting called black-box prompt learning, where we only have access to the output of prediction APIs without the need to access the PLM's parameters or gradients. The black-box prompt is optimized without the requirements of tuning pre-trained models, saving the fine-tuning costs.

• We propose a new black-box discrete prompt learning (BDPL) method to solve this new problem, and demonstrate its effectiveness in dealing with domain shifts on various tasks.

• We conduct comprehensive analyses on eleven benchmark datasets under cloud-device collaboration settings, demonstrating its effectiveness for commercial APIs. BDPL has a much wider range of applications than previous methods, such as transfer learning, model personalization, and decentralized training.

## 2 Approach

In our setting, the input is a sentence $S = s_1 s_2 \cdots s_l \cdots s_m$ with $s_l$ indicating the $l$-th token and the output corresponds to its category $y$. Our goal is to learn $n$ discrete prompt tokens $T = t_1 t_2 \cdots t_i \cdots t_n = \mathcal{V}[j_1]\mathcal{V}[j_2] \cdots \mathcal{V}[j_i] \cdots \mathcal{V}[j_n]$, which are prepended to the input sentence to create the user query $[T, S]$. Note that $\mathcal{V}$ represents the vocabulary list consisting of a total of $N$ tokens, and $t_i = \mathcal{V}[j_i]$ is the $i$-th token in $T$ and $j_i$-th token in $\mathcal{V}$.

The overall architecture is shown in Figure 1. During the black-box training, we freeze the prediction model $\mathcal{G}$ with a stop-gradient strategy, and only optimize the discrete prompts $T$. Here, we assume independent categorical distribution for each prompt index $j_i \sim \mathrm{Cat}(\boldsymbol{p}_i)$, where the random variable $j_i$ is sampled with the probability distribution $\boldsymbol{p}_i = [p_{i,1}, \cdots, p_{i,N}]$ over the $N$ token indexes, where $\boldsymbol{p}_i \in \mathcal{C}$ and $\mathcal{C} = \{\boldsymbol{p} : \|\boldsymbol{p}\|_1 = 1, 0 \preceq \boldsymbol{p} \preceq 1\}$. Since $\boldsymbol{p}_i$ is independent of each other, the joint probability of the whole discrete prompt is $P(T) = \Pi_{i=1}^n P(t_i) = \Pi_{i=1}^n p_{i,j_i}$.

Because the prediction model's parameters are invisible and gradients cannot be back-propagated to the prompts, it is no longer possible to directly update the prompts by back-propagating through $\nabla_{\boldsymbol{p}_i}\mathcal{L}(\mathcal{G}([T, S], y))$, where $y$ is the label. Inspired by the policy gradient algorithm in discrete optimization, we resort to estimating the gradients without back-propagation to accomplish **black-box** training.

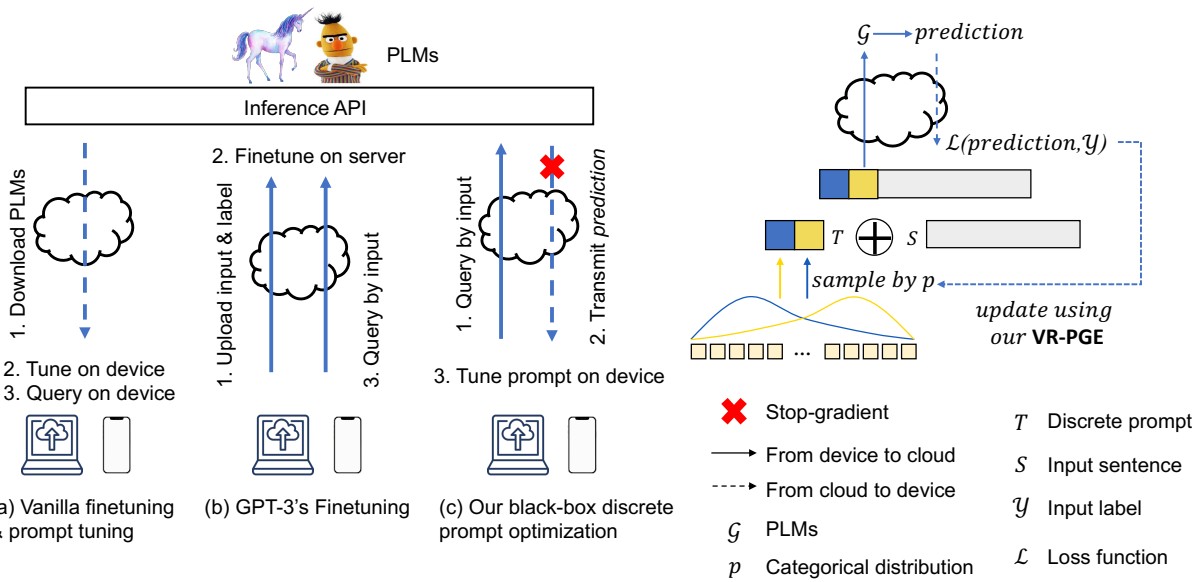

Figure 1: Schematic illustrations of the comparisons across various tuning paradigms and the cloud-device interplay at the training phase of our algorithm. **Left:** (a) Vanilla finetuning and prompt tuning can be conducted on user devices in a white-box manner since the PLMs are feasible to be duplicated at user devices. After tuning, users can still access services of PLMs on the device. (b) Increasing scale hinders the democratizing of PLMs. In GPT-3's finetuning, users have to upload the input and associated labels to the server. After finetuning, the model is saved on the server. (c) In our black-box discrete prompt learning setting, users send queries to the server and then rely on PLMs' predictions to update their discrete prompts on the devices using a gradient-free optimization. **Right:** In our framework, the user query is created by concatenating the discrete prompt and the input sentence, where the prompt tokens are sampled from their categorical distributions respectively. After calculating the loss between the PLMs' predictions and input labels, we apply a variance-reduced policy gradient algorithm to estimate the gradient of the categorical distribution and update it accordingly.

For abbreviation, we denote the $\mathcal{L}(\mathcal{G}([T,S], y))$ as $\mathcal{L}(T)$ since $S, y$ can be deemed as constants here. By the virtue of the policy gradient estimator (PGE), we can optimize the loss function via forward propagation with:

$$\mathbb{E}_T\left[\mathcal{L}(T)\right] = \int \mathcal{L}(T)P(T)\,\mathrm{d}T, \tag{1}$$

and estimate the gradient of $\boldsymbol{p}_i$ by:

$$
\begin{aligned}
\nabla_{\boldsymbol{p}_i}\mathbb{E}_T\left[\mathcal{L}(T)\right] &= \int \mathcal{L}(T)\nabla_{\boldsymbol{p}_i}P(T)\,\mathrm{d}T \\
&= \int \mathcal{L}(T)\frac{P(T)}{P(T)}\nabla_{\boldsymbol{p}_i}P(T)\,\mathrm{d}T \\
&= \int P(T)\mathcal{L}(T)\nabla_{\boldsymbol{p}_i}\log P(T)\,\mathrm{d}T \\
&= \mathbb{E}_{P(T)}\left[\mathcal{L}(T)\nabla_{\boldsymbol{p}_i}\log \Pi_{j=1}^n P(t_j)\right] \\
&= \mathbb{E}_{P(T)}\left[\mathcal{L}(T)\nabla_{\boldsymbol{p}_i}\log P(t_i)\right]
\end{aligned}
\tag{2}
$$

The $j$-th component of $\nabla_{\boldsymbol{p}_i}\log P(t_i)$ could be solved explicitly by:

$$\nabla_{p_{i,j}}\log P(t_i) = \nabla_{p_{i,j}}\log p_{i,j_i} \tag{3}$$

---
**Algorithm 1** The black-box discrete optimization procedures.

---
**Require:** Input batch $S$, Label batch $Y$, Parameter of categorical distribution $\boldsymbol{p}_1, \cdots, \boldsymbol{p}_n$, Prediction model $\mathcal{G}$, Loss function $\mathcal{L}$.

1: **for** $k \leq I$ **do**
2:      Sample $j_1^{(k)} \sim \mathrm{Cat}(\boldsymbol{p}_1), \cdots, j_n^{(k)} \sim \mathrm{Cat}(\boldsymbol{p}_n)$
3:      $T^{(k)} = t_1^{(k)} \cdots t_n^{(k)} = \mathcal{V}[j_1^{(k)}] \cdots \mathcal{V}[j_n^{(k)}]$
4: **end for**
5: $\mathcal{L}_{\mathrm{avg}} = \frac{1}{I} \sum_{k=1}^{I} \mathcal{L}(\mathcal{G}[T^{(k)}, S], Y)$
6: **for** $i \leq n$ **do**
7:      $\boldsymbol{g}_{\boldsymbol{p}_i}^{vr} = \frac{1}{I-1} \sum_{k=1}^{I} \nabla_{\boldsymbol{p}_i} \log P(t_i^{(k)})(\mathcal{L}(\mathcal{G}[T^{(k)}, S], Y) - \mathcal{L}_{\mathrm{avg}})$
8:      $\boldsymbol{p}_i \leftarrow \mathrm{proj}_{\mathcal{C}}(\boldsymbol{p}_i - \eta \cdot \boldsymbol{g}_{\boldsymbol{p}_i}^{vr})$
9: **end for**
10: **return** $\boldsymbol{p}_1, \cdots \boldsymbol{p}_n$

---

When $j = j_i$, it is obvious that $\nabla_{p_{i,j}} \log P(t_i) = \frac{1}{p_{i,j_i}}$. When $j \neq j_i$, equation (3) is calculated by:

$$
\begin{aligned}
\nabla_{p_{i,j}} \log P(t_i) &= \nabla_{p_{i,j}} \log\left(1 - \sum_{k=1, k \neq j_i}^{N} p_{i,k}\right) \\
&= -\frac{1}{1 - \sum_{k=1, k \neq j_i}^{N} p_{i,k}} \\
&= -\frac{1}{p_{i,j_i}}
\end{aligned}
\tag{4}
$$

However, consistent with previous policy gradient applications (Sutton et al., 1999; Rezende et al., 2014; Jang et al., 2017; Zhou et al., 2021), we observed that conventional PGE suffers from high variance, which makes it challenging to converge in practice. Therefore, we adopted a variance-reduced policy gradient estimator (VR-PGE) as described in Williams (1992); Dong et al. (2020); Zhou et al. (2021). The estimated gradient is calculated by:

$$
\boldsymbol{g}_{\boldsymbol{p}_i}^{vr} = \frac{1}{I-1} \sum_{k=1}^{I} \left( \mathcal{L}(T^{(k)}) - \frac{1}{I} \sum_{j=1}^{I} \mathcal{L}(T^{(j)}) \right) \nabla_{\boldsymbol{p}_i} \log P(t_i)
\tag{5}
$$

where $T^{(k)}, k = 1, \cdots, I$ are sampled independently from $P(T)$.

Thus, the prompt token distribution $\boldsymbol{p}_i$ can be updated by a projected stochastic gradient descent algorithm:

$$
\boldsymbol{p}_i \leftarrow \mathrm{proj}_{\mathcal{C}}(\boldsymbol{p}_i - \eta \cdot \boldsymbol{g}_{\boldsymbol{p}_i}^{vr}), i = 1, \cdots, n
\tag{6}
$$

where $\eta$ is the learning rate of prompt learning, $I$ is the sample size, and $\mathrm{proj}_{\mathcal{C}}$ is the projection calculation (details are presented in the Appendix).

Here we introduce the detailed training procedure for updating the prompts using our proposed VR-PGE, whose mini-batch version is displayed in Algorithm 1. Assuming the input data is divided into $B$ batches, and within each batch, we will perform $I$ iterations of sampling to reduce the variance of estimation. Specifically, at the $k$-th iteration within each batch, we first sample the sequence of prompt tokens $T^{(k)} = \mathcal{V}[j_1^{(k)}]\mathcal{V}[j_2^{(k)}] \cdots \mathcal{V}[j_n^{(k)}]$ according to the joint distribution $P(T)$. When $T^{(k)}$ is created, we will prepend it to the input sentence $S$ and feed the query $[T^{(k)}, S]$ into the black-box pre-trained language model $\mathcal{G}$, which will return back the prediction. In light of the model prediction and ground-truth label $Y$, we then calculate the loss $\mathcal{L}(\mathcal{G}[T^{(k)}, S], Y)$. Then the estimated gradients $\boldsymbol{g}_{\boldsymbol{p}_i}^{vr}$ for each $p_i$ could be obtained by executing Equation (5) after sampling all $I$ prompt sequences for the training batch. Finally, the categorical distributions are updated by Equation (6).

**Vocabulary Construction** A natural question is how to construct the vocabulary $V$ and what is the size $N$. Inspired by the observation in Diao et al. (2021), which revealed the importance of domain-specific and

task-specific words and ngrams in representation learning, we introduce such important ngrams as prompt candidates. Therefore, we adopt pointwise mutual information (PMI) to construct the vocabulary of candidate prompt tokens in an unsupervised way. For each sentence in the training set, we calculate the PMI by

$$\text{PMI}(\bar{x}, \widetilde{x}) = \log \frac{p(\bar{x}\widetilde{x})}{p(\bar{x})p(\widetilde{x})}, \tag{7}$$

where $\bar{x}$ and $\widetilde{x}$ are two adjacent words in the sentence, and $p(x)$ is the probability of an n-gram $x$. If the PMI score between these two adjacent words is high, they have a high probability of co-occurrence and are more likely to form an n-gram, suggesting they are good collocation pairs. If the PMI score is lower than a threshold $\sigma$, a delimiter is inserted between $\bar{x}$ and $\widetilde{x}$. As a result, the sentence will be segmented by several delimiters. Finally, we obtain a list of ngrams $V$ by extracting those consecutive words after segmentation and with a frequency of at least $f$. As for the size $N$, we observe that large $N$ will cause an unstable optimization process and even divergence. Therefore, we keep $N$ between 50 and 200.

## 3 Experimental Settings

In this section, we first introduce the datasets and evaluation metrics (§3.1), followed by the baseline models (§3.2). Lastly, we describe the implementation details (§3.3).

### 3.1 Datasets and Evaluation Metrics

In order to examine the model's ability in generic classification tasks as well as domain-specific classification tasks, we include seven datasets from the GLUE benchmark (Wang et al., 2019): MNLI (Williams et al., 2018), QQP (Iyer et al., 2017), SST-2 (Socher et al., 2013), MRPC (Dolan & Brockett, 2005), CoLA (Warstadt et al., 2019), QNLI (Wang et al., 2019), RTE (Dagan et al., 2005; Haim et al., 2006; Giampiccolo et al., 2007; Bentivogli et al., 2009), and four domain-specific datasets: CITATIONINTENT (Jurgens et al., 2018), SCIERC (Luan et al., 2018), RCT (Dernoncourt & Lee, 2017), HYPERPARTISAN (Kiesel et al., 2019) from specific domains including computer science, biomedical science and news following Gururangan et al. (2020); Diao et al. (2021). The statistics of these datasets are shown in Table 2. Considering the data sparsity issue and large query costs[4] in cloud-device collaboration, we conduct our experiments on a popular and more realistic setting — few-shot learning, where huge models have shown their powerful ability (Brown et al., 2020). We follow Perez et al. (2021) to simulate a true $k$-shot learning setting. We randomly sample $k$ data from the original training set for each class to construct the training set and another different $k$ data to construct the validation set. The original validation set will be used as the test set. Because the size of the QQP and RCT validation sets is too large, we randomly sample 1K data to save costs. We adopt Matthews Correlation Coefficient for CoLA, F1-score for QQP, MRPC, CITATIONINTENT, SCIERC, HYPERPARTISAN, RCT, and accuracy for SST-2, RTE, QNLI, SST-2, IMDB, CR, MR, and MPQA following Wang et al. (2019); Diao et al. (2021). MNLI results are an average of MNLI-match and MNLI-mismatch accuracy.

### 3.2 Baselines

For GPT-3-based models, because previous white-box tuning methods and black-box continuous prompt tuning methods (e.g., BBT) cannot be applied to GPT-3, we compare our model with the following baselines.

- **GPT-3's FineTuning**[5]: a GPT-3 inference API that is fine-tuned entirely on a labeled dataset (black-box).
- **ManualPrompt**: a GPT-3 inference API with manually composed prompts to conduct the zero-shot evaluation. The human-written prompts are shown in Appendix A.5 (black-box).
- **InContextLearning** (Brown et al., 2020): a GPT-3 inference API with a set of examples containing training sentences and labels as the input prefix, which is then prepended to the input texts (black-box).

For RoBERTa-based models, we adopt the RoBERTa-large as the backbone and the following tuning methods.

- **Vanilla FineTuning** (Liu et al., 2019): a RoBERTa-large model that is fine-tuned entirely on a labeled dataset (white-box).

---

[4]For example, only one training epoch on 10, 000 sentences with 300, 000 tokens will cause 6 USD costs for GPT-3-Davinci. Not to mention tens of hundreds of rounds of training.

[5]https://beta.openai.com/docs/guides/fine-tuning

Table 2: The statistics of seven datasets in the generic domain and four datasets in the specific domain. CI, SE, HP denote CITATIONINTENT, SCIERC, HYPERPARTISAN, respectively. |**L**|: number of classes for classification tasks. Note that we sample the few-shot training split and development split from the original training split for few-shot setting as described in Section 3.1.

| Dataset | \|**L**\| | \|**Train**\| | \|**Dev**\| | \|**Test**\| | Type | Metrics | Domain |
|---|---|---|---|---|---|---|---|
| *Generic Tasks* | | | | | | | |
| MNLI | 3 | 393K | 9.8K | 9.8K | NLI | acc. | fiction, reports |
| QQP | 2 | 364K | 40K | 391K | paraphrase | F1 | Quora |
| SST-2 | 2 | 6.7K | 872 | 1.8K | sentiment | acc. | movie reviews |
| MRPC | 2 | 3.7K | 408 | 1.7K | paraphrase | F1 | news |
| CoLA | 2 | 8.6K | 1K | 1K | acceptability | Matthews corr. | books, articles |
| QNLI | 2 | 105K | 5.5K | 5.5K | NLI | acc. | Wikipedia |
| RTE | 2 | 2.5K | 277 | 3K | NLI | acc. | news, Wikipedia |
| *Domain-Specific Tasks* | | | | | | | |
| CI | 6 | 1.6K | 114 | 139 | citation intent | F1 | computer science |
| SE | 7 | 3.2K | 455 | 974 | relation classification | F1 | computer science |
| RCT | 5 | 180K | 30K | 30K | abstract sentence roles | F1 | biomedical |
| HP | 2 | 516 | 64 | 65 | review helpfulness | F1 | reviews |

- **PromptTuning** (Lester et al., 2021): a frozen RoBERTa-large model with continuous prompt embeddings prepended to the input, and learned by gradients (white-box).
- **P-Tuning v2** (Liu et al., 2021a): a frozen RoBERTa-large model with continuous prompt embeddings prepended to each layer, and learned by gradients (white-box).
- **AutoPrompt** (Shin et al., 2020): a frozen RoBERTa-large model with discrete prompts optimized based on gradient-guided search (white-box).
- **FeatureProbe** (Peters et al., 2019): a frozen RoBERTa-large model outputs the features given inputs and a newly added classification layer is trained with the gradients (white-box).
- **ManualPrompt**: a frozen RoBERTa-large model with manually composed prompts to conduct the zero-shot evaluation. The human-written prompts are shown in Appendix A.5 (black-box).
- **InContextLearning** (Brown et al., 2020): a frozen RoBERTa-large model with a set of examples containing training sentences and labels as the input prefix, which is then prepended to the input texts (black-box).
- **BBT** (Sun et al., 2022): a frozen RoBERTa-large model with continuous prompts that are optimized by covariance matrix adaptation evolution strategy (black-box).
- **RLPrompt** (Deng et al., 2022): a frozen RoBERTa-large model with discrete prompts that are generated by a policy network and optimized by a reward function (black-box).

## 3.3 Implementation

For GPT-3 experiments, we conduct experiments with four variants: GPT-3-Ada, GPT-3-Babbage, GPT-3-Curie, and GPT-3-Davinci. The batch size of training and evaluation is set to 4 to fulfill the query length limit (i.e., 2048). We call the APIs directly from OpenAI's services[6]. For RoBERTa-large experiments, we initialize it with pre-trained weights by Huggingface's Transformers library[7]. The batch size of training and evaluation is set to 16 and 32, respectively. The number of API calls is limited to 8000 across all datasets.

For BDPL, we optimize the prompts by AdamW (Loshchilov & Hutter, 2019) for 30 epochs with a learning rate of $1 \times 10^{-4}$. The prompt length is 50, and the size of the candidate prompt list $N$ is 100. Other hyper-parameters are detailed in the Appendix A.3.

---

[6]https://openai.com/api/
[7]https://github.com/huggingface/transformers

Table 3: The overall performance of black-box prompt and the comparison on eleven datasets with GPT-3. We report average scores across three random seeds, with standard deviations as subscripts. Avg. denotes the average score across all tasks. $Cost denotes the money cost in US dollars for calling GPT-3's API during training and inference. FT: GPT-3's FineTuning. MP: ManualPrompt. ICL: InContextLearning.

| Dataset | MNLI | QQP | SST-2 | MRPC | CoLA | QNLI | RTE | CI | SE | RCT | HP | Avg. | $Cost |
|---|---|---|---|---|---|---|---|---|---|---|---|---|---|
| *GPT-3 Ada* | | | | | | | | | | | | | |
| FT | $38.5_{0.8}$ | $44.5_{1.4}$ | $71.6_{1.2}$ | $45.7_{4.3}$ | $0.0_{0.0}$ | $49.8_{2.1}$ | $52.7_{1.2}$ | $27.7_{3.2}$ | $3.5_{0.3}$ | $57.0_{4.2}$ | $24.4_{1.4}$ | 37.8 | 5.6 |
| MP | $26.5_{0.9}$ | $31.2_{1.8}$ | $63.1_{1.3}$ | $35.6_{2.1}$ | $0.0_{0.0}$ | $45.6_{1.5}$ | $47.3_{2.0}$ | $26.9_{2.0}$ | $1.2_{0.6}$ | $15.8_{2.4}$ | $15.2_{1.8}$ | 28.0 | 0.5 |
| ICL | $36.3_{0.7}$ | $40.3_{1.3}$ | $64.6_{0.8}$ | $40.5_{1.5}$ | $1.3_{1.4}$ | $48.8_{1.7}$ | $48.7_{1.3}$ | $28.2_{0.3}$ | $2.5_{0.4}$ | $22.7_{2.6}$ | $20.0_{2.3}$ | 32.2 | 5.7 |
| BDPL | $37.1_{0.7}$ | $45.1_{0.5}$ | $68.8_{0.9}$ | $43.2_{2.5}$ | $2.0_{0.4}$ | $51.2_{0.4}$ | $52.7_{1.3}$ | $28.3_{1.5}$ | $3.8_{0.3}$ | $45.7_{2.9}$ | $22.4_{1.7}$ | 36.4 | 3.2 |
| *GPT-3 Babbage* | | | | | | | | | | | | | |
| FT | $40.7_{0.5}$ | $46.2_{1.4}$ | $87.4_{1.5}$ | $66.4_{1.7}$ | $0.3_{0.1}$ | $50.9_{0.2}$ | $52.3_{1.0}$ | $5.2_{0.4}$ | $4.1_{1.0}$ | $61.1_{5.2}$ | $33.3_{1.3}$ | 40.7 | 8.5 |
| MP | $28.9_{0.8}$ | $34.1_{1.2}$ | $83.5_{1.2}$ | $62.4_{3.2}$ | $0.2_{0.1}$ | $48.8_{1.4}$ | $51.2_{0.6}$ | $31.4_{2.8}$ | $1.7_{0.5}$ | $21.7_{2.3}$ | $27.2_{1.5}$ | 35.6 | 0.6 |
| ICL | $35.7_{0.9}$ | $45.2_{1.9}$ | $86.2_{1.4}$ | $65.4_{1.7}$ | $2.6_{0.0}$ | $48.3_{0.9}$ | $51.5_{0.4}$ | $13.1_{1.5}$ | $2.5_{0.9}$ | $36.7_{1.8}$ | $32.2_{1.4}$ | 38.1 | 7.1 |
| BDPL | $41.0_{0.6}$ | $50.4_{1.5}$ | $86.4_{1.1}$ | $67.7_{1.2}$ | $2.8_{0.1}$ | $52.1_{0.3}$ | $53.1_{1.0}$ | $40.2_{2.5}$ | $3.2_{0.8}$ | $45.2_{2.2}$ | $30.4_{2.3}$ | 43.0 | 4.0 |
| *GPT-3 Curie* | | | | | | | | | | | | | |
| FT | $42.2_{2.8}$ | $53.3_{1.4}$ | $88.9_{3.1}$ | $76.3_{2.1}$ | $3.4_{1.3}$ | $49.0_{1.3}$ | $54.5_{1.7}$ | $28.4_{1.9}$ | $5.1_{0.8}$ | $50.6_{1.3}$ | $43.3_{1.5}$ | 45.0 | 42.3 |
| MP | $34.5_{1.9}$ | $44.3_{2.1}$ | $84.2_{1.4}$ | $73.3_{1.2}$ | $2.0_{0.6}$ | $47.2_{0.9}$ | $44.0_{1.2}$ | $19.2_{1.3}$ | $2.8_{0.3}$ | $31.0_{1.4}$ | $37.1_{1.5}$ | 38.1 | 2.5 |
| ICL | $38.0_{2.1}$ | $47.2_{2.3}$ | $87.0_{1.9}$ | $81.0_{1.3}$ | $2.8_{0.0}$ | $46.8_{1.1}$ | $46.2_{1.6}$ | $15.2_{1.9}$ | $4.8_{1.3}$ | $50.1_{2.3}$ | $39.0_{2.3}$ | 41.6 | 28.5 |
| BDPL | $42.5_{1.9}$ | $52.0_{1.5}$ | $88.0_{2.2}$ | $82.6_{1.1}$ | $4.0_{1.3}$ | $50.1_{0.8}$ | $55.8_{1.3}$ | $25.5_{1.8}$ | $3.4_{1.8}$ | $49.6_{2.4}$ | $39.4_{1.6}$ | 44.8 | 16.2 |
| *GPT-3 Davinci* | | | | | | | | | | | | | |
| FT | $60.2_{3.8}$ | $67.8_{2.1}$ | $92.9_{2.4}$ | $84.6_{1.3}$ | $55.3_{1.5}$ | $54.2_{2.5}$ | $57.0_{1.2}$ | $35.4_{1.7}$ | $10.3_{2.3}$ | $51.6_{2.7}$ | $60.1_{1.8}$ | 57.2 | 423.2 |
| MP | $40.2_{2.5}$ | $39.2_{1.6}$ | $86.7_{2.7}$ | $69.7_{2.1}$ | $55.2_{2.4}$ | $28.0_{1.3}$ | $55.3_{1.9}$ | $25.6_{2.0}$ | $4.9_{1.0}$ | $26.8_{1.9}$ | $52.6_{1.4}$ | 44.0 | 25.0 |
| ICL | $52.7_{2.9}$ | $55.6_{3.4}$ | $87.2_{3.3}$ | $82.4_{1.7}$ | $56.7_{2.0}$ | $17.9_{1.7}$ | $56.6_{2.3}$ | $30.1_{3.0}$ | $9.2_{1.5}$ | $44.4_{2.2}$ | $55.4_{2.5}$ | 49.8 | 206.1 |
| BDPL | $54.6_{2.4}$ | $57.8_{2.1}$ | $89.3_{3.0}$ | $83.4_{1.4}$ | $58.4_{1.4}$ | $56.2_{1.5}$ | $57.2_{0.8}$ | $34.6_{2.0}$ | $6.6_{2.1}$ | $48.8_{2.5}$ | $58.5_{2.4}$ | 55.0 | 161.5 |

## 4 Experimental Results

The overall results on eleven datasets are reported in Tables 3 and 4.

We first verify our proposed method's effectiveness on a purely black-box setting with GPT-3 APIs. From Table 3, BDPL shows great superiority across eleven datasets. Compared with ManualPrompt and InContextLearning, BDPL demonstrates significant improvements, which are 8.35% and 4.35% on average of Ada, Babbage, Curie, and Davinci models. BDPL also achieves comparable performance with GPT-3's fine-tuning, which requires large money costs and uploading user's data. In Babbage, BDPL even outperforms GPT-3's fine-tuning. As the experiments are conducted on the few-shot setting, where a small number of data are available to fine-tune the models' parameters, for large models like GPT-3, overfitting could be a serious problem that deteriorates the performance so that fine-tuning is inferior to BDPL. Although careful adjustment of the fine-tuning algorithm may mitigate overfitting and improve accuracy, it needs a lot of manual effort and money costs, which is implausible for cloud-device collaboration. Moreover, it is observed that ManualPrompt and InContextLearning with less capable versions of GPT-3 (e.g., Ada and Babbage) fail in some challenging datasets (e.g., CoLA and SE) but BDPL performs well on them. With the increase of the model capacity (from Ada to Davinci), we observed ManualPrompt and InContextLearning could also solve them, which is consistent with recent observations of large model's emergent abilities (Wei et al., 2022). From this perspective, our method offers another option in addition to increasing model size, which is an efficient solution for less capable models to perform challenging tasks.

Table 4: The overall performance of black-box prompt and the comparison on eleven datasets with RoBERTa-large. We report average scores across three random seeds, with standard deviations as subscripts. Avg. denotes the average score across all tasks. FT: Vanilla FineTuning. ICT: InContextLearning.

| DATASET | MNLI | QQP | SST-2 | MRPC | CoLA | QNLI | RTE | CI | SE | RCT | HP | Avg. |
|---|---|---|---|---|---|---|---|---|---|---|---|---|
| *White-Box Methods* | | | | | | | | | | | | |
| FT | $50.8_{1.2}$ | $60.8_{1.9}$ | $86.5_{2.0}$ | $78.4_{1.3}$ | $20.4_{1.9}$ | $53.2_{1.8}$ | $55.6_{2.5}$ | $37.4_{1.7}$ | $23.1_{1.6}$ | $45.2_{5.2}$ | $55.5_{2.3}$ | 51.5 |
| PromptTuning | $36.5_{0.9}$ | $50.2_{1.5}$ | $70.7_{2.6}$ | $52.7_{3.4}$ | $8.0_{0.7}$ | $53.5_{1.6}$ | $56.3_{1.6}$ | $34.4_{2.6}$ | $28.6_{2.5}$ | $36.7_{3.1}$ | $47.4_{3.2}$ | 43.2 |
| P-Tuning v2 | $44.2_{1.7}$ | $57.4_{2.4}$ | $80.4_{1.2}$ | $62.4_{2.0}$ | $8.9_{2.7}$ | $51.5_{1.3}$ | $53.1_{1.7}$ | $31.4_{4.2}$ | $24.6_{2.5}$ | $35.4_{3.9}$ | $55.4_{4.2}$ | 45.9 |
| AutoPrompt | $40.1_{1.5}$ | $45.7_{1.3}$ | $71.5_{2.1}$ | $63.8_{3.1}$ | $5.4_{2.3}$ | $50.2_{1.3}$ | $52.1_{1.6}$ | $27.9_{2.9}$ | $21.5_{2.5}$ | $29.6_{2.5}$ | $40.6_{3.8}$ | 40.8 |
| FeatureProbe | $46.5_{1.8}$ | $56.3_{1.1}$ | $79.5_{1.6}$ | $68.9_{1.7}$ | $15.6_{1.2}$ | $50.5_{0.2}$ | $54.1_{2.5}$ | $22.3_{2.0}$ | $20.8_{3.6}$ | $31.2_{4.7}$ | $60.1_{2.6}$ | 46.0 |
| *Black-Box Methods* | | | | | | | | | | | | |
| ManualPrompt | $35.9_{1.3}$ | $49.8_{0.9}$ | $77.2_{2.1}$ | $70.4_{1.6}$ | $0.6_{0.0}$ | $49.2_{1.1}$ | $48.2_{0.6}$ | $12.3_{2.4}$ | $9.6_{1.4}$ | $11.7_{1.5}$ | $35.7_{1.6}$ | 36.4 |
| ICT | $37.2_{1.6}$ | $50.1_{0.9}$ | $82.8_{2.1}$ | $72.1_{2.3}$ | $1.1_{0.4}$ | $50.8_{0.5}$ | $49.3_{2.3}$ | $14.6_{1.7}$ | $9.2_{1.5}$ | $25.8_{1.6}$ | $38.5_{2.4}$ | 39.2 |
| BBT | $40.6_{2.5}$ | $55.2_{3.1}$ | $85.3_{3.9}$ | $66.4_{3.7}$ | $5.5_{2.7}$ | $55.4_{3.2}$ | $52.6_{2.2}$ | $17.4_{5.4}$ | $16.4_{0.9}$ | $31.7_{1.5}$ | $47.2_{4.8}$ | 43.1 |
| RLPrompt | $42.8_{3.2}$ | $53.7_{2.2}$ | $88.4_{1.9}$ | $68.9_{2.1}$ | $5.0_{1.1}$ | $52.6_{1.4}$ | $51.8_{1.8}$ | $19.2_{3.3}$ | $18.8_{1.5}$ | $30.1_{2.7}$ | $44.9_{2.4}$ | 43.3 |
| BDPL | $42.5_{1.8}$ | $56.4_{1.9}$ | $87.6_{2.1}$ | $78.1_{3.7}$ | $4.6_{1.2}$ | $53.1_{1.1}$ | $53.5_{0.9}$ | $24.0_{1.3}$ | $21.5_{2.0}$ | $36.6_{3.2}$ | $45.6_{3.4}$ | 45.8 |

In addition to the auto-regressive model, we also conduct additional experiments with an encode-only model, RoBERTa-large. Because the weights of RoBERTa-large are released and gradients can be leveraged, several white-box baseline models are introduced for comparison. First, our model outperforms all black-box methods, demonstrating the effectiveness of our proposed black-box discrete prompt optimization. Second, BDPL achieves comparable performance compared with white-box prompt-based methods including both discrete and continuous prompts. It is observed that BDPL even outperforms some white-box methods (e.g., PromptTuning and AutoPrompt). We attribute this phenomenon to the overfitting of white-box methods in terms of the given few-shot examples while BDPL does not suffer severe overfitting due to its exploration mechanism. We perform an ablation study in Section E to reveal the effect of data size and accuracy. Given that white-box prompt tuning methods cannot be applied in black-box settings when the gradients are unavailable, previous black-box methods such as InContextLearning and BBT can achieve 2.82% and 6.66% improvement on average over ManualPrompt. BDPL outperforms the previous black-box methods BBT and RLPrompt by an average of 2.7% and 2.5%, respectively. Compared with BBT, our method, BDPL, not only outperforms it but is also more practical considering its discrete nature. BBT is optimizing continuous prompts and cannot be directly fed into current prediction APIs. We also notice that there is still a large gap between FineTuning and all other methods. FineTuning updates the full model with gradients and huge parameters, serving as an upper bound for all methods. Across eleven tasks, it is observed that the BDPL on domain-specific datasets is as effective as on generic datasets. While it is known that domain shift introduced difficulty for models to deal with, BDPL offers an effective solution to domain-specific datasets.

## 5 Analysis

We analyze several aspects of BDPL, including the effects of different training data sizes, prompt lengths, training budgets, and learning objectives. In addition, we examine the transferability of our learned prompts under the transfer learning setting and the explanation of prompts. We choose GPT-3 Babbage as the backbone model in the following discussion. The details are illustrated in this section.

### 5.1 Ablation Study

**Effects of Training Data Size** First, we analyze the effects brought by four different training data sizes: 4-shot, 8-shot, 16-shot, and 32-shot. Experiments are conducted on MRPC and RCT datasets. As shown in the left part of Figure 2 (a) and (b), with the increase in training data size, the performance of FT, InContextLearning, and BDPL is improved on both MRPC and RCT, which is consistent with the assumption that more data brings sufficient training. Compared with baseline models, our model achieved consistent

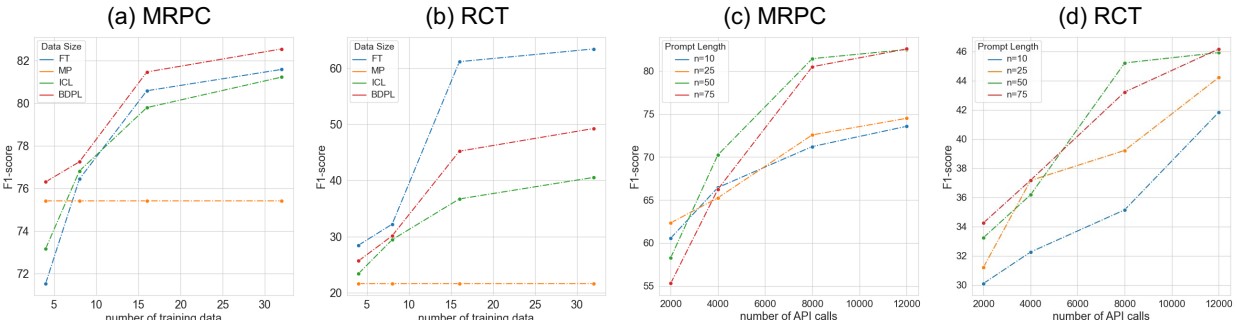

Figure 2: The effects of training data size, prompt length, and the number of API calls on MRPC and RCT. FT, MP, and ICL denote GPT-3's FineTuning, ManualPrompt, and InContextLearning, respectively.

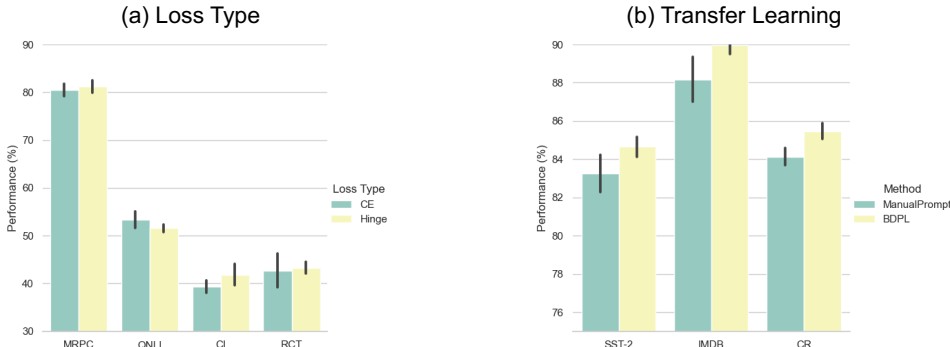

Figure 3: **(a) Ablations of loss function.** CE and Hinge represent cross-entropy loss and hinge loss, respectively. **(b) Transfer learning performance.** SST-2 is the source task while IMDB and CR are two tasks in the target domain.

improvement over ManualPrompt and InContextLearning, verifying its effectiveness and scalability under different data sizes.

**Effects of Prompt Length** It is known that prompt-based methods are sensitive to many aspects of prompts, including contexts (Jiang et al., 2020), orders (Lu et al., 2022) and lengths (Lester et al., 2021), and inappropriately designed prompts lead to bad performance. Here we study the effects of different prompt lengths on MRPC and RCT. Considering the maximum number of tokens is 2048 for the input of GPT-3 API, too many prompt tokens (e.g., more than 100) cause additional costs and even failure of queries. Therefore, we conduct experiments with length $= 10, 25, 50,$ and 75. The results are shown in the Figure 2 (c) and (d). With the increase in prompt length, the performance increases at first and decreases when the prompt length reaches 75 in most cases. We conclude that the approximate best prompt length is 50, since a shorter prompt length limits the representation capacity while a longer prompt length might involve noises contained in the training data and is hard to optimize.

**Effects of Training Budgets** Training budgets are essential factors for efficient-tuning methods. An efficient method is expected to achieve good performance with as few as training budgets. We measure the budgets by the number of prediction API calls and report the performance of our models with different numbers of API calls in Figure 2 (c) and (d). It is observed that with the increase of API calls, BDPL under different settings obtains performance gains because of sufficient training. All settings could converge within 12, 000 API calls. We also find that the 50-prompt performs well at first but the gap between 75-prompt and 50-prompt narrows down when the training budget grows, and finally, 75-prompt achieves competitive performance. It told us that if we do not have a sufficient training budget, it would be better to apply fewer prompt tokens, which are easier to optimize.

**Effects of Different Objectives** In the previous experiments, the prompts are optimized with the cross-entropy loss, and here we explore the effectiveness of our model with different objectives. With the same

| Task | Prompt + Input | Prediction | Label |
|------|----------------|------------|-------|
| CoLA |  Our friends won't buy this analysis, let alone the next one we propose .  | Unacceptable | Acceptable |
| | as time game second family group company take full at way only  Our friends **won't** buy this analysis, **let alone** the next one we **propose** .  | Acceptable | |
| RTE |  one of the dead was a child, said a doctor at Civil Hospital Karachi.   A doctor was killed by his parents .  | Entailment | Not Entailment |
| | got because people during go N or work both support come also  one of the dead was a child, **said** a **doctor** at Civil Hospital Karachi.   A **doctor** was **killed** by his parents .  | Not Entailment | |
| CI |  This appeared to solve the problem, and the results presented later for the average degree of generalisation do not show an over-generalisation compared with those given in Li and Abe ( 1998 ) .  | Background | CompareOr Contrast |
| | last ie may man life show F best most state well around  This appeared to solve the problem, and the **results** presented later for the **average** degree of generalisation do not show an over-generalisation **compared** with those given in Li and Abe ( 1998 ) .  | CompareOr Contrast | |
| RCT |  It is not clear whether these patients would benefit from antifungal treatment . | Results | Background |
| | go such time part event city use found season play news people  It is not **clear** whether these patients would **benefit** from antifungal **treatment** . | Background | |

Figure 4: Four correctly predicted examples by BDPL. We display the prompts and salience map of the token ``. The prompts are in green and the input tokens are in red. The salient tokens are highlighted in a blue background, where the darker color denotes the more dominant weights for the prediction.

setting as our main experiment, we conduct further experiments with hinge loss on four datasets: MRPC, QNLI, CI, and RCT. We find that our model with both objectives can achieve comparable results. As shown in Figure 3 (a), the model with hinge loss outperforms that with cross-entropy loss on MRPC, CI, and RCT, but underperforms it on QNLI. On average, our approach with hinge loss works as well as that with cross-entropy loss. It is flexible enough to work with different objectives, and we hope to extrapolate to any kind of human-designed objectives.

**Effects of Transfer Learning** A critical advantage of discrete prompts is the possibility of transferring prompts learned from one task to another because discrete tokens share the same text space instead of specific latent space for continuous prompts. To verify the transferability of black-box optimized prompt tokens, we conduct experiments on three sentiment analysis datasets (*i.e.*, SST-2 (Socher et al., 2013), IMDB (Maas et al., 2011), and CR (Hu & Liu, 2004)) with GPT-3-Babbage model in the 16-shot setting. First, we use SST-2 as the source task following Vu et al. (2022) and optimize the discrete prompt tokens with our proposed BDPL. Then we obtain those selected prompt tokens and simply prepend them to the beginning of the input of the target task, IMDB and CR. Our setting assumes no training data in the target domain, so we directly test the performance in the target task. Following Wang et al. (2021a), for CR, we randomly sample 2,000 instances as the test set. The results are shown in Figure 3 (b). Consistent with the previous observation in Section 4, BDPL outperforms ManualPrompt in the source task by a large margin. Moreover, learned prompts are helpful in two target tasks, demonstrating that our black-box method is robust under transfer learning settings. The experimental results display the expansion potential of *prompt transfer*, which is a promising practical application of BDPL, especially when there are $N$ edge devices sharing a similar task, but they have no training data. We can update the black-box prompts in a general domain and then transfer them to the target domain.

## 5.2 Prompt Explanation

To intuitively understand the prompts, we visualize the salience maps using the Language Interpretability Tool (LIT) (Tenney et al., 2020). We choose CoLA, RTE, CI, and RCT datasets because the sentences contained in these datasets are easy for human interpretation. The comparisons between models with discrete prompts and without them are shown in Figure 4. By adding discrete prompt tokens, the model is able to find coherence-related clues. For example, CoLA aims to distinguish the acceptability of sentences. The grammar-related word 'won't' and phrase 'let alone' dominate its prediction, which is consistent with the decision process of human beings. Similar observations are found in other datasets. Due to space limitations, more visualized examples are not shown here. Based on considerable empirical evidence, we conclude that BDPL can capture helpful information to guide the model.

We notice that most of the optimized prompts are readable but in-comprehensible, useful to model improvement, but semantically confusing to humans. Ilyas et al. (2019b) find that neural networks rely on some *non-robust* features to achieve the highest possible accuracy. These *non-robust* features are usually semantically meaningless to humans, but we can still train a well-performed model only using these features. We argue that while the optimized prompts that our method finds are meaningless to humans, they are useful for the models to make a more accurate prediction. In contrast, forcing the prompts to have semantic meaning may remove the useful information and leads to degraded performance. This observation is consistent with previous discrete prompt learning studies (Shin et al., 2020).

## 6 Related Work

In this section, we present the review on prompt learning for pre-trained language models and black-box optimization.

### 6.1 Prompts for Pre-trained Models

Large pre-trained language models are of great importance and a standard paradigm is pre-training a language model on a large unlabeled corpus and then fine-tuning the pre-trained model on different supervised tasks. This approach shows great improvements on lots of downstream tasks but it needs lots of computational resources to change all the parameters and has to save a copy for each task. Therefore, prompt-based learning, which does not require tuning the large model, is proposed to solve the problem. Based on the format of prompts, the prompt-based learning can be categorized into two kinds: discrete prompt (Wallace et al., 2019; Shin et al., 2020; Jiang et al., 2020; Gao et al., 2021; Ben-David et al., 2022) and continuous prompt (Zhong et al., 2021; Qin & Eisner, 2021; Hambardzumyan et al., 2021; Liu et al., 2021b; Han et al., 2021; Li & Liang, 2021). The discrete prompt is usually a sequence of tokens or natural language phrases while the continuous prompt is designed as a sequence of vectors. However, all of these studies are limited to a white-box setting, which requires accessing all the parameters of a pre-trained model so that the gradients could be back-propagated to optimize the prompts. Recently, Sun et al. (2022) proposed black-box tuning methods but they are optimizing continuous prompts, which is impractical in real applications, because most of the commercial APIs do not accept continuous vectors as input. Our method, black-box prompt learning, provides a truly black-box solution with discrete optimization, which optimizes a set of discrete prompts without accessing the pre-trained model. There are some concurrent works (Deng et al., 2022; Hou et al., 2022) exploring black-box tuning methods for large language models. PromptBoosting (Hou et al., 2022) ensembles a large number of weak learners by the AdaBoost algorithm to pair pre-generated prompts with different elements of the LM's output distribution. They learn the verbalizer instead of discrete prompts of the language model, which is complementary to BDPL. RLPrompt (Deng et al., 2022) generates discrete prompts by a policy network and optimizes it by a reward function. In contrast, we apply a variance-reduced policy gradient estimator to optimize a few independent categorical probabilities to select the appropriate prompt tokens. In addition, we are the first to show that the black-box prompt learning methods can generalize to real-world large language models like GPT-3.

### 6.2 Black-Box Optimization

One of the applications of black-box optimization is the score-based black-box adversarial attack (Ilyas et al., 2018; 2019a; Huang & Zhang, 2020; Andriushchenko et al., 2020; Cheng et al., 2019), where the models are also invisible to the attacker. These studies use zeroth-order optimization methods such as natural evolution strategy (NES) (Wierstra et al., 2014) to optimize the input and increase the loss to fool the model. Instead of deteriorating the models' performance in the adversarial attack, our direction is to find the inputs that improve the accuracy. Policy gradient (Sutton et al., 1999), which also belongs to the black-box optimization, is widely used in reinforcement learning to find the best policy. In contrast to NES that can only be used to search in the continuous search, policy gradient allows the choice of discrete policy and can be used to find the optimal discrete prompts. BDPL uses black-box optimization methods to find the optimal prompts, which is a novel application direction of these methods.

Another line of research to adapt the black-box model is knowledge distillation (KD) (Hinton et al., 2015), which learns a student model with the outputs of large models. KD can be used to learn black-box models (Nguyen et al., 2022; Wang, 2021), perform domain adaptation (Liang et al., 2022), and adversarially attack the model Zhang et al. (2022). Despite the wide applications of black-box KD, learning the models with KD still requires a large number of queries and data for training the student network. On the contrary, our proposed approach, BDPL, is much more lightweight, and only needs to train a few prompts with a small amount of data. Moreover, under the scenario of cloud-device collaboration, BDPL only needs negligible computation on the edge devices while KD has to train the local student network with large computational resources. Therefore, BDPL is more practical for the scenario of cloud-device collaboration.

## 7  Conclusion

This paper proposes a novel setting for text categorization namely black-box prompt learning, where a large pre-trained model is invisible so that the gradients cannot be back-propagated to update the prompts. Compared with the standard pre-training then fine-tuning paradigm, our approach only requires updating very few parameters. Compared with previous prompt-based methods, our approach does not require the visibility of pre-trained models, and thus it provides more flexibility in practical applications. We propose a black-box prompt learning method, BDPL, which employs a variance-reduced policy gradient estimator to approximate the gradients, and then update the prompts. Experimental results demonstrate that our approach outperforms all black-box methods and is comparable with white-box methods, illustrating the effectiveness of black-box optimization. Experiments on the transfer learning settings further show the potential of our approach in realistic scenarios, where the pre-trained model is deployed on the cloud, and the prompt learning can be implemented on each device.

In the future, we would like to explore the effectiveness of our proposed methods on more commercial classifiers, such as Google Cloud APIs, Microsoft Azure APIs and so on. The black-box prompt learning for large multi-modal models (Wang et al., 2021b; Singh et al., 2022; Wang et al., 2021c; Zhou et al., 2022; Wang et al., 2022; Diao et al., 2023) is another important scenario to explore in future work.

## Acknowledgments

We thank the anonymous reviewers for their valuable suggestions. This work was supported by the General Research Fund (GRF) of Hong Kong (No. 16310222 and No. 16201320). Shizhe Diao, Ruijia Xu, and Yong Lin were supported by the Hong Kong Ph.D. Fellowship Scheme (HKPFS).

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

## A    Implementation Details

### A.1    Computing Infrastructure

For experiments on GPT-3, we directly call its APIs without any GPU for computation. For experiments on RoBERTa, they are conducted with NVIDIA 2080Ti GPUs with 11GB memory.

### A.2    Evaluation Measures

For tasks from the GLUE Benchmark, we adopt Matthews correlation coefficient for CoLA, F1 for MRPC, and accuracy for RTE and QNLI following their original metric choices. We adopt macro-F1 for CITATIONINTENT, SciERC, RCT, and HYPERPARTISAN as evaluation metrics.

### A.3    Bounds of Hyper-parameters

Table 5: Bounds of hyper-parameters.

| Hyper-parameter | GPT-3 | RoBERTa |
|---|---|---|
| number of epochs | 30 | 30 |
| train batch size | 4 | 32 |
| eval and test batch size | 4 | 16 |
| prompt length | $\{10, 12, 25, 50, 75\}$ | |
| learning rate | [1e-5, 1e-3] | |
| dropout | 0.1 | |
| learning rate optimizer | AdamW | |
| loss type | {hinge loss, cross-entropy loss} | |

### A.4    Configuration of Best Model

The configuration of the best model for each dataset is shown in Table 6.

Table 6: Configuration of the best model for each dataset. The rest hyper-parameters are the default value in Table 5.

| DATASET | MNLI | QQP | SST-2 | MRPC | CoLA | QNLI | RTE | CI | SE | RCT | HP |
|---|---|---|---|---|---|---|---|---|---|---|---|
| prompt length | 10 | 25 | 50 | 50 | 50 | 50 | 50 | 50 | 50 | 50 | 50 |
| learning rate | 2e-4 | 1e-4 | 2e-4 | 1e-4 | 3e-4 | 2e-4 | 1e-4 | 1e-4 | 1e-4 | 1e-4 | 1e-4 |

### A.5    Manual Templates

The manual templates are shown in Table 7.

## B    Projection Calculation

The projection from $\boldsymbol{z}$ to $\mathcal{C}$ can be calculated by:

---
**Algorithm 2** Projection from $\boldsymbol{z}$ to $\mathcal{C}$

---
**Require:** a vector $\boldsymbol{z}$.
 1: Solve $v_1$ from $\mathbf{1}^\top [\min(1, \max(0, \boldsymbol{z} - v_1^* \mathbf{1}))] - 1 = 0$.
 2: $\boldsymbol{p} \leftarrow \min(1, \max(0, \boldsymbol{p} - v_1^* \mathbf{1}))$.
**output** $\boldsymbol{p}$

---

Table 7: Prompts and label descriptions of ManualPrompt method. Most of them are from Gao et al. (2021).

| Dataset | Template |
|---|---|
| MNLI | sentence$_1$ entailment?[MASK], sentence$_2$. (yes/no) |
| QQP | sentence$_1$ ?[MASK], sentence$_2$. (yes/no) |
| SST-2 | sentence$_1$. It was [MASK]. (great/terrible) |
| MRPC | sentence$_1$ ?[MASK], sentence$_2$. (yes/no) |
| CoLA | sentence$_1$. correct? [MASK]. (yes/no) |
| QNLI | sentence$_1$ entailment?[MASK], sentence$_2$. (yes/no) |
| RTE | sentence$_1$ entailment?[MASK], sentence$_2$. (yes/no) |
| CI | sentence$_1$. What is the intent?[MASK]. (background/compare/extends/future/motivation/uses) |
| SE | sentence$_1$. What is the relation?[MASK]. (compare/conjunction/evaluate/feature/ hyponym/part/used) |
| RCT | sentence$_1$. It is [MASK]. (background/conclusion/method/objective/result) |
| HP | sentence$_1$. It is [MASK]. (yes/no) |
| IMDB | sentence$_1$. It was [MASK]. (great/terrible) |
| CR | sentence$_1$. It was [MASK]. (great/terrible) |

*Proof.* The projection from $\boldsymbol{z}$ to set $\mathcal{C}$ can be formulated in the following optimization problem:

$$\min_{\boldsymbol{p} \in \mathbb{R}^n} \frac{1}{2} \|\boldsymbol{p} - \boldsymbol{z}\|^2,$$

$$s.t. \mathbf{1}^\top \boldsymbol{p} = 1 \text{ and } 0 \le p_i \le 1.$$

Then we solve the problem with the Lagrangian multiplier method.

$$L(\boldsymbol{p}, v) = \frac{1}{2} \|\boldsymbol{p} - \boldsymbol{z}\|^2 + v(\mathbf{1}^\top \boldsymbol{p} - 1) \tag{8}$$

$$= \frac{1}{2} \|\boldsymbol{p} - (\boldsymbol{z} - v\mathbf{1})\|^2 + v(\mathbf{1}^\top \boldsymbol{z} - 1) - \frac{n}{2}v^2. \tag{9}$$

with $0 \le p_i \le 1$. Minimize the problem with respect to $\boldsymbol{p}$, we have

$$\tilde{\boldsymbol{p}} = \mathbf{1}_{\boldsymbol{z} - v\mathbf{1} \ge 1} + (\boldsymbol{z} - v\mathbf{1})_{1 > \boldsymbol{z} - v\mathbf{1} > 0} \tag{10}$$

Then we have

$$
\begin{aligned}
g(v) =& L(\tilde{\boldsymbol{p}}, v) \\
=& \frac{1}{2} \|[\boldsymbol{z} - v\mathbf{1}]_- + [\boldsymbol{z} - (v+1)\mathbf{1}]_+\|^2 \\
& + v(\mathbf{1}^\top \boldsymbol{z} - 1) - \frac{n}{2}v^2 \\
=& \frac{1}{2} \|[\boldsymbol{z} - v\mathbf{1}]_-\|^2 + \frac{1}{2} \|[\boldsymbol{z} - (v+1)\mathbf{1}]_+\|^2 \\
& + v(\mathbf{1}^\top \boldsymbol{z} - 1) - \frac{n}{2}v^2. \\
g'(v) =& \mathbf{1}^\top [v\mathbf{1} - \boldsymbol{z}]_+ + \mathbf{1}^\top [(v+1)\mathbf{1} - \boldsymbol{z}]_- \\
& + (\mathbf{1}^T \boldsymbol{z} - 1) - nv \\
=& \mathbf{1}^\top \min(1, \max(0, \boldsymbol{z} - v\mathbf{1})) - 1.
\end{aligned}
$$

It is easy to verify that $g'(v)$ is a monotone decreasing function with respect to $v$ and we can use a bisection method solve the equation $g'(v) = 0$ with solution $v_1^*$. Finally we have

$$\boldsymbol{p}^* = \mathbf{1}_{\boldsymbol{z} - v_1^* \mathbf{1} \ge 1} + (\boldsymbol{z} - v_1^* \mathbf{1})_{1 > \boldsymbol{z} - v_1^* \mathbf{1} > 0} \tag{11}$$

$$= \min(1, \max(0, \boldsymbol{z} - v_1^* \mathbf{1})). \tag{12}$$

□

Table 8: The effects of different prompt token positions. We report average scores across three random seeds, with standard deviations as subscripts. Avg. denotes the average score across all tasks. FT: GPT-3's FineTuning. MP: ManualPrompt. ICL: InContextLearning.

| Dataset | MNLI | QQP | SST-2 | MRPC | CoLA | QNLI | RTE | CI | SE | RCT | HP | Avg. |
|---|---|---|---|---|---|---|---|---|---|---|---|---|
| *GPT-3 Babbage* | | | | | | | | | | | | |
| FT | $40.7_{0.5}$ | $46.2_{1.4}$ | $87.4_{1.5}$ | $66.4_{1.7}$ | $0.3_{0.1}$ | $50.9_{0.2}$ | $52.3_{1.0}$ | $5.2_{0.4}$ | $4.1_{1.0}$ | $61.1_{5.2}$ | $33.3_{1.3}$ | 40.7 |
| MP | $28.9_{0.8}$ | $34.1_{1.2}$ | $83.5_{1.2}$ | $62.4_{3.2}$ | $0.2_{1.0}$ | $48.8_{1.4}$ | $51.2_{0.6}$ | $31.4_{2.8}$ | $1.7_{0.5}$ | $21.7_{2.3}$ | $27.2_{1.5}$ | 35.6 |
| ICL | $35.7_{0.9}$ | $45.2_{1.9}$ | $86.2_{1.4}$ | $65.4_{1.7}$ | $2.6_{0.0}$ | $48.3_{0.9}$ | $51.5_{0.4}$ | $13.1_{1.5}$ | $2.5_{0.9}$ | $36.7_{1.8}$ | $32.2_{1.4}$ | 38.1 |
| BDPL | $41.0_{0.6}$ | $50.4_{1.5}$ | $86.4_{1.1}$ | $67.7_{1.2}$ | $2.8_{0.1}$ | $52.1_{0.3}$ | $53.1_{1.0}$ | $40.2_{2.5}$ | $3.2_{0.8}$ | $45.2_{2.2}$ | $30.4_{2.3}$ | 43.0 |
| BDPL-infix | $25.1_{1.1}$ | $35.6_{1.8}$ | $80.2_{2.3}$ | $60.3_{1.4}$ | $0.5_{0.2}$ | $50.8_{0.3}$ | $51.2_{1.0}$ | $15.2_{2.9}$ | $2.0_{0.7}$ | $10.5_{0.9}$ | $20.2_{1.7}$ | 32.0 |
| BDPL-suffix | $39.1_{1.7}$ | $47.5_{2.1}$ | $85.6_{3.1}$ | $64.2_{2.6}$ | $1.1_{0.4}$ | $51.8_{1.6}$ | $52.9_{1.2}$ | $23.7_{1.6}$ | $2.9_{0.7}$ | $43.1_{2.3}$ | $28.8_{1.2}$ | 40.1 |

## C  Effects of Prompt Positions

In our main experiments, we follow existing prompt-based learning studies (Li & Liang, 2021) to prepend some prompt tokens to the original sequence. We also investigate the effects of prompt positions. First, we introduce two new baselines based on GPT-3-babbage: suffix-tuning (placing prompt tokens after the original sequence) and infix-tuning (placing prompt tokens in the middle of the sequence). The results are shown in 8. From the results, we observed that prefix-tuning outperforms suffix-tuning and infix-tuning by a large margin. We attribute this performance drop to the position embedding of the learned prompts. Compared with infix-tuning and suffix-tuning, prefix-tuning keeps the prompt tokens at the same position, which is consistent during the learning process. However, the infix-tuning and suffix-tuning require the prompts to have adapting ability to dynamic positions. In addition, putting the prompt tokens in the middle of the sequence may break the semantic meaning of the original sequence, leading to even worse results than ManualPrompt. These observations are consistent with the prefix-tuning (Li & Liang, 2021), so we decide to adopt prefix-tuning in our main method.

## D  Case Studies

More case studies for prompt explanation are provided in Figure 5.

## E  Performance with More Data

It is not very straightforward that black-box methods can outperform the white–box methods (i.e., Prompt-Tuning and AutoPrompt). However, some similar observations were reported in previous and concurrent black-box studies. For example, BBT outperforms PromptTuning and AutoPrompt, and RLPrompt outperforms AutoPrompt. According to our experiments, we attribute this phenomenon to the overfitting of white-box methods in terms of the given few-shot examples. First, in our experiments, we found that PromptTuning and AutoPrompt have lower training losses. It suggests that white-box methods tend to overfit the small training data (our experiments are 16-shot). Furthermore, we gradually increase the number of training data and verify whether the experimental results are consistent with our conjecture. Specifically, we increase the training data from 16-shot to 32-shot, 64-shot, and 128-shot. The results are shown in Figure 6. It is observed that as the amount of training data increases, the white-box methods will outperform black-box methods, which demonstrates that white-box methods are better than black-box methods given sufficient data. Based on these experiments, we believe black-box methods are good at few-shot settings due to their exploration mechanism, which can mitigate overfitting.

| Task | Prompt + Input | Prediction | Label |
|------|----------------|------------|-------|
| CoLA |  Our friends won't buy this analysis, let alone the next one we propose .  | Unacceptable | Acceptable |
| | any boy are some me were go any as said about one read out eat he be on for the you last could was and John more his never got like man would as book on will did who it gave can time saw about girl sent hit in he  Our friends **won't** buy this analysis, **let alone** the next one we **propose** .  | Acceptable | |
| RTE |  one of the dead was a child, said a doctor at Civil Hospital Karachi.   A doctor was killed by his parents .  | Entailment | Not Entailment |
| | found would by million had in from of than died being years made because including three her one said New will government announced company South political United more other people under been for many could new will died from against million being because would under former first including died will  one of the **dead** was a **child**, **said** a **doctor** at Civil Hospital Karachi.   A **doctor** was **killed** by his parents .  | Not Entailment | |
| CI |  This appeared to solve the problem, and the results presented later for the average degree of generalisation do not show an over-generalisation compared with those given in Li and Abe ( 1998 ) .  | Background | CompareOr Contrast |
| | such word can between used use example learning information analysis proposed described parsing structure models translation results semantic Collins lexical between rules proposed similar syntactic such which system learning grammar translation text language word training work information learning used described structure learning lexical analysis between structure results information syntactic parsing  This appeared to solve the problem, and the **results** presented later for the **average** degree of generalisation do not show an over-generalisation **compared** with those given in Li and Abe ( 1998 ) .  | CompareOr Contrast | |
| RCT |  It is not clear whether these patients would benefit from antifungal treatment . | Results | Background |
| | group treatment study placebo was during clinical significantly after weeks significant of primary baseline during placebo clinical trial randomized groups was therapy risk symptoms significantly group of patients was two changes versus therapy more use days all vs also blood participants p We To who that months was and as  It is not **clear** whether these patients would **benefit** **from** antifungal **treatment** . | Background | |

Figure 5: Four correctly predicted examples by BDPL. We display the prompts and salience map of the token ``. The prompts are in green and the input tokens are in red. The salient tokens are highlighted in a blue background, where the darker color denotes the more dominant weights for the prediction.

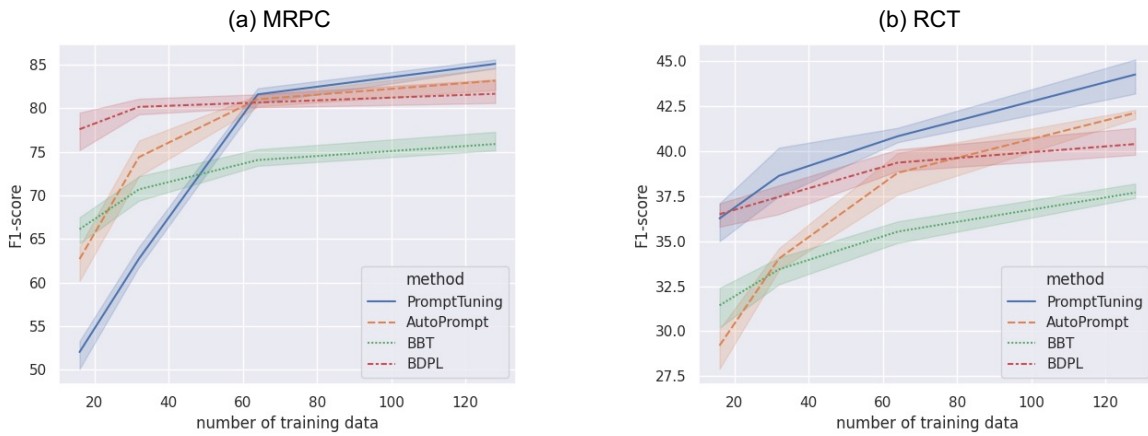

Figure 6: The effects of training data size on MRPC and RCT with RoBERTa-large model. PromptTuning and AutoPrompt are two white-box methods. BBT and BDPL are two black-box methods.

