# OpenReview forum: "Black-Box Prompt Learning for Pre-trained Language Models"
_TMLR — Accepted by TMLR_

### Review · Reviewer_v5DN · 2022-12-04

**Summary Of Contributions:**

This paper introduces a black-box prompt learning method, where some generated discrete prompt tokens are prepended to the original sequence to improve the quality of the generated sequences. The learning procedure leverages a no back-propagation based approach. The model is evaluated on various tasks and datasets and shows an advanced performance boost.

**Audience:**

Yes

**Broader Impact Concerns:**

Not applicable.

**Claims And Evidence:**

Yes

**Requested Changes:**

- Provide some justification for the algorithmic design decisions of the proposed methods.

- Release the code for review if possible.

**Strengths And Weaknesses:**

Strengths:

- This is a very interesting paper that introduces a novel discrete prompt-learning method, this method enables a more efficient and secure way of applying large-scale language models.

- The empirical study is very solid. The proposed methods are compared with various alternatives under different tasks and datasets. The performance boost is significant.


Weakness:

- In general, I just have some small concerns about some design decisions without justification. For example, why the learned prompt tokens are placed before the original sequences? Alternatives could be to place them after the original sequence, or insert some learned tokens between the original sequences. I am not suggesting the proposed method is inappropriate, I am just curious if such alternatives have been considered, if so, why they did not work.

---

> ### Author Response · Authors · 2022-12-19
> **Response to Reviewer v5DN**
>
> Dear Reviewer v5DN,
>
> Thank you very much for your thorough review and insightful feedback! We address your comments one by one as follows:
>
> **[Justification for the Algorithmic Design]**
>
> Thanks for your question!
> In our main experiments, we follow existing prompt-based learning studies (e.g., Prefix-tuning) to prepend some prompt tokens to the original sequence. We agree that providing more justification for our algorithm design is helpful, so we have added a new section to investigate the effects of different positions. Specifically, we introduced two new baselines based on GPT-3-babbage: suffix-tuning (placing the prompt tokens after the original sequence) and infix-tuning (placing the prompt tokens in the middle of the sequence). The results of these new baselines are shown below:
>
>
> | Model \ Dataset  | MNLI | QQP | SST-2 | MRPC | CoLA | QNLI | RTE | CI | SE | RCT | HP | Avg. |
> | ----------- | ----------- | ----------- | ----------- | ----------- | ----------- | ----------- | ----------- | ----------- | ----------- | ----------- | ----------- | ----------- |
> | FT | 40.7 | 46.2 | 87.4 | 66.4 | 0.3 | 50.9 | 52.3 | 5.2 | 4.1 | 61.1 | 33.3 | 40.7 |
> | MP | 28.9 | 34.1 | 83.5 | 62.4 | 0.2 | 48.8 | 51.2 | 31.4 | 1.7 | 21.7 | 27.2 | 35.6 |
> | ICL | 35.7 | 45.2 | 86.2 | 65.4 | 2.6 | 48.3 | 51.5 | 13.1 | 2.5 | 36.7 | 32.2 | 38.1 |
> | BDPL-prefix (ours) | 41.0 | 50.4 | 86.4 | 67.7 | 2.8 | 52.1 | 53.1 | 40.2 | 3.2 | 45.2 | 30.4 | 43.0 |
> | BDPL-infix | 25.1 | 35.6 | 80.2 | 60.3 | 0.5 | 50.8 | 51.2 | 15.2 | 2.0 | 10.5 | 20.2 | 32.0 |
> | BDPL-suffix | 39.1 | 47.5 | 85.6 | 64.2 | 1.1 | 51.8 | 52.9 | 23.7 | 2.9 | 43.1 | 28.8 | 40.1 |
>
>
> From the results, we observed that prefix-tuning outperforms suffix-tuning and infix-tuning by a large margin. We attribute this to the position embedding of the learned prompts. Compared with infix-tuning and suffix-tuning, prefix-tuning keeps the prompt tokens at the same position, which is consistent during the learning process. However, infix-tuning and suffix-tuning require the prompts to have the ability to adapt to dynamic positions. In addition, placing the prompt tokens in the middle of the sequence may break the semantic meaning of the original sequence, resulting in even worse performance than ManualPrompt.
> These observations are consistent with the prefix-tuning paper [1], so we decided to adopt prefix-tuning in our main method.
>
> Thank you very much for your constructive suggestion! We have included the details of these experiments in Appendix C of our revised paper.
>
>
>
> **[Code Release]**
>
> Thanks for your question! We have uploaded our code to the system for your review. We plan to make it available to the public in the near future.
>
>
> [1] Li, X. L., & Liang, P. (2021, August). Prefix-Tuning: Optimizing Continuous Prompts for Generation. In Proceedings of the 59th Annual Meeting of the Association for Computational Linguistics and the 11th International Joint Conference on Natural Language Processing (Volume 1: Long Papers) (pp. 4582-4597).

---

### Review · Reviewer_k32w · 2022-12-15

**Summary Of Contributions:**

This paper looks into the problem of adapting pretrained models to downstream tasks through prompt learning and without the access to the model weights and gradient information, the so called black-box setting where pre-trained LMs are exposed through only inference APIs. In particular, the paper looks into the black-box prompt learning scenario, where the end devices can  query the outputs of the pre-trained model and use that to optimize prompts locally. Finally, the paper focuses on discrete prompts learning because existing cloud LM prediction APIs only accept discrete prompts as inputs. Under this setup, the paper formulates this black-box prompt learning as a discrete token selection problem and optimizes a categorical distribution via a policy gradient algorithm (gradient-free). Furthermore, the paper also introduces a variance-reduced policy gradient estimator to reduce the high variance issue in policy gradient optimization. Evaluation on RoBERTa and GPT-3 APIs show that the proposed method can effectively and efficiently adapt pretrained models through black-box manner.

**Audience:**

Yes

**Claims And Evidence:**

Yes

**Requested Changes:**

The paper overall looks into a very interesting problem and makes a good attempt of resolving the problem that may lead to an impactful direction. There are a few comments the work:

1. While the proposed black-box prompt learning is indeed an interesting problem, the paper does not discuss too much about alternative methods for adapting pre-trained LMs without accessing model parameters. For example, it might be worthwhile discussing how the proposed method relates to knowledge distillation which also learns a separate model by querying the outputs of the teacher model (in this case, the pre-trained PLM). In particular, KD also does not require the client to have access to model weights/gradients. The resulting student model can be used for adaptation. Do we really need to formulate a new problem when existing techniques might already help resolve the problem?

2. It is unclear why the paper only uses a subset of the GLUE datasets for evaluation, e.g., MNLI, QQP, SST-2, and STSB in GLUE  are not included. This raises questions on whether the reported datasets are cherry-picked. To be more convincing, it would be better to report the remaining GLUE task results.

3. The paper compares its approach with white-box fine-tuning methods, and it seems that the black-box prompt learning methods are getting similar accuracy as the white-box fine-tuning method in several cases. For example, BDPL is getting <1 point difference and sometimes better performance on MRPC, QNLI, RTE on both GPT-3 and RoBERTa. There are two issues related to this evaluation results: i) the fine-tuning results for RoBERTa on these tasks are noticeably worse by a large margin than their originally reported scores (e.g., please refer to Table 5 in RoBERTa paper and GLUE leaderboard). (ii) MRPC, CoLA, RTE have large variances that are hard to interpret their results without confidence intervals. To be more convincing, the authors might want to explain why the RoBERTa fine-tuning results on those GLUE tasks are much worse than the original paper and include the other GLUE tasks together with standard deviation to the results.

**Strengths And Weaknesses:**

Strengths:
- The paper introduces the black-box prompt learning problem, which to the best of the reviewer's knowledge, is a new and interesting problem.
- The paper takes practical constraints into consideration, such as the cloud LM APIs only taking discrete tokens and no easy access to model weights/gradients into consideration when designing and evaluating their approach.
- A comprehensive evaluation of different tuning methods (10 methods).
- The paper is clearly written and easy to follow.

Weaknesses:
- Lack of discussion with alternative methods for black-box adaptation.
- Evaluation datasets appear to be cherry-picked.
- Lack of explanation on why some baseline results are noticeably worse than their original reported results.

---

> ### Author Response · Authors · 2022-12-19
> **Response to Reviewer k32w**
>
> Dear Reviewer k32w,
>
> Thank you very much for your comprehensive review and valuable suggestions! We appreciate your detailed comments and have addressed them in our revised paper as follows:
>
> **[BDPL v.s. Knowledge Distillation]**
>
> As you mentioned, knowledge distillation (KD) could learn a student model using the outputs of large language models (e.g., GPT-3). However, our approach has several differences from KD:
>
> 1. KD requires a large amount of data to learn the full decision boundary of the large models and achieve similar performance as large models, which can be costly in terms of computation and queries. In contrast, our proposed approach is much more lightweight and only needs to train a few prompts instead of a student network.
>
> 2. Moreover, it is more practical to use large language models in a few-shot setting because the amount of labeled data may be limited. In such cases, it can be challenging to learn a student network with KD.
>
> 3. In terms of training, our framework is a single-branch approach where the input is only forwarded once to the teacher network in the cloud while KD usually employs a double-branch architecture, requiring feeding the input into both the cloud teacher and local student networks, respectively. The training of the local student network can be computationally expensive, which is unaffordable for small edge devices like smartphones. In contrast, our approach only requires minimal computation on the edge devices, making it more suitable for cloud-device collaboration.
>
> 4. In terms of inference, KD stores the student networks locally and performs inference on the edge devices, while our approach performs inference in the cloud to utilize the computational resources of the cloud to help small edge devices.
>
>
> **[GLUE Benchmark]**
>
> We wanted to cover a diverse set of datasets, so we randomly picked four domain-agnostic datasets from GLUE and four domain-specific datasets from previous studies. In the original submission, we reported the result of the SST-2 dataset in Section 5.1 Effects of Transfer Learning (Figure 3 (b)).
> We agree that including the other GLUE tasks is helpful in demonstrating the model’s performance. Therefore, we added the results of MNLI, QQP, SST-2 in Table 3 and Table 4 of our revised paper.
> We adopted the common setting in few-shot classification studies [1,2], so we did not include STSB and WNLI. STSB is a regression task, and WNLI has adversarial distributions between train/dev/test splits (https://gluebenchmark.com/faq). Thus, these metrics cannot accurately reflect the real effect.
>
>
> **[RoBERTa Performance]**
>
> The reason for the gap between the performance of RoBERTa paper and ours is that RoBERTa is trained on the full datasets while we are working in a few-shot setting. Given the limited data, it is expected that the performance would drop. Similar to other few-shot classification studies [1,2,3,4], we conducted our experiments in this setting for several reasons.
>
> 1. In many real-world scenarios, the amount of available data may be limited and it can be costly to collect more data.
>
> 2. For large language models (e.g., GPT-3, OPT, BLOOM-175B), we aim to achieve good performance using a small amount of data, so that these models can be used more efficiently, which is also in line with previous work on in-context learning.
>
> 3. Even if we have large data, it is costly to query GPT-3. In Table 3, we display the money cost, where fine-tuning a model on GPT-3 Davinci costs more than 280 USD.
>
> 4. It is difficult to compare with baseline models (i.e., in-context learning) because GPT-3 only accepts a sequence with fewer than 2048 tokens. Providing too many examples cannot improve the performance of in-context learning.
>
> We hope that these explanations address your concerns and we look forward to your further feedback. Thank you again for your review.
>
> [1] Sun, T., Shao, Y., Qian, H., Huang, X., & Qiu, X. (2022). Black-box tuning for language-model-as-a-service. In Proceedings of 39th International Conference on Machine Learning.
>
> [2] Schick, T., & Schütze, H. (2021, April). Exploiting Cloze-Questions for Few-Shot Text Classification and Natural Language Inference. In Proceedings of the 16th Conference of the European Chapter of the Association for Computational Linguistics: Main Volume (pp. 255-269).
>
> [3] Brown, T., Mann, B., Ryder, N., Subbiah, M., Kaplan, J. D., Dhariwal, P., ... & Amodei, D. (2020). Language models are few-shot learners. Advances in neural information processing systems, 33, 1877-1901.
>
> [4] Gao, T., Fisch, A., & Chen, D. (2021, August). Making Pre-trained Language Models Better Few-shot Learners. In Proceedings of the 59th Annual Meeting of the Association for Computational Linguistics and the 11th International Joint Conference on Natural Language Processing (Volume 1: Long Papers) (pp. 3816-3830).

---

### Review · Reviewer_u59W · 2022-12-22

**Summary Of Contributions:**

The paper introduces a method called Black-box Discrete Prompt Learning (BDPL) for adapting Pre-trained Language Models (PLMs) for different downstream tasks in a cloud-device collaboration setting. BDPL optimizes PLMs through prompt learning, which adjusts a few discrete parameters rather than fine-tuning the entire model. BDPL was tested on RoBERTa and GPT-3 and showed significant improvement on multiple benchmarks, with case studies also conducted to analyze the method under various conditions.

**Audience:**

Yes

**Broader Impact Concerns:**

The black-box setting of BDPL is preferable for protecting the cloud infrastructure from potential attacks and misuse, which could have a positive impact on security and reliability.

**Claims And Evidence:**

No

**Requested Changes:**

Could you provide a plausible (and theoretical) explanation of how black-box prompt learning could perform better than white-box one?

Can you compare other prompt learning methods with reinforcement learning [1, 2]?
[1] Deng et al., RLPrompt: Optimizing Discrete Text Prompts With Reinforcement Learning
[2] Hou et al., PromptBoosting: Black-Box Text Classification with Ten Forward Passes

What’s the difference between ManualPrompt and InContextLearning?

The prompt length N looks large (50~200). Isn’t it expensive? The length of the prompts in Figure 4 is short. why?

Page 7: first InContextLearning -> GPT-3 not RoBERTa-large

Question: how do you initialize prompt tokens? From random tokens?

**Strengths And Weaknesses:**

As explained, the black-box prompt learning setting could be more realistic when we can only access large language models’ outputs without other information such as gradients. However, it doesn’t make sense how and why black-box prompt learning could outperform white-box prompt learning in terms of accuracy and cost because it is a more restrictive setting. Moreover, I wonder whether compared baselines are not strong enough.

---

> ### Author Response · Authors · 2022-12-31
> **Response to Reviewer u59W**
>
> Dear Reviewer u59W,
>
> Thank you very much for your thorough review and insightful feedback! We address your comments one by one as follows:
>
> **[Black-box v.s. White-box]**
>
> We agree with the reviewer that it is not very straightforward that black-box methods can outperform the white–box methods (i.e., PromptTuning and AutoPrompt). However, we want to mention that some similar observations were reported in previous and concurrent black-box studies. For example, BBT outperforms PromptTuning and AutoPrompt, and RLPrompt outperforms AutoPrompt.
>
> According to our experiments, we attribute this phenomenon to the overfitting of white-box methods in terms of the given few-shot examples. First, in our experiments, we found that PromptTuning and AutoPrompt have lower training losses. It suggests that white-box methods tend to overfit the small training data (our experiments are 16-shot). Furthermore, we gradually increase the number of training data and verify whether the experimental results are consistent with our conjecture. Specifically, we increase the training data from 16-shot to 32-shot, 64-shot, and 128-shot. The results are shown in Appendix E and Figure 6. It is observed that as the amount of training data increases, the white-box methods will outperform black-box methods, which demonstrates that white-box methods are better than black-box methods given sufficient data.
> However, in the few-shot setting, the black-box methods are able to better mitigate overfitting due to their exploration mechanism.
>
>
>
> **[Comparison with New Methods]**
>
> We have followed the suggestion of the reviewer and included RLPrompt as one of the compared methods. The experimental results are shown in Table 4 of our revised paper, where our proposed method outperforms RLPrompt by an average of 2.5%. In terms of PromptBoosting, it was submitted to arXiv on Dec. 19, which is more than one month after we submitted this work to TMLR on Nov. 8. However, we agree with the reviewer that this concurrent work is also promising to discuss. To this end, we have cited and discussed it in our revised paper. Thanks again for your suggestion.
>
>
> **[ManualPrompt v.s. InContextLearning]**
>
> The main difference between ManualPrompt and InContextLearning is whether using additional data as examples (also called demonstrations). ManualPrompt only uses a few prompt tokens (templates + label words) without any examples while InContextLearning uses additional examples.
>
> **[Prompt Length]**
>
> In Section 5.1 and Figure 2, we have conducted an ablation study to verify the results in terms of prompt length varying from 10 to 75 and found that the empirically optimal prompt length is 50. Because the vocabulary size is about 100, the search space is relatively small and the corresponding cost is acceptable, which is reported in Table 3 of our paper. As for the prompt length in Figure 4, the main motivation of this case study is to understand the optimized prompts by visualizing the salience maps. Due to space limitations, we simply displayed the cases with short prompts, whose performance is good as well. We also conducted such exploration with 50 prompts and obtained the same conclusion as reported in Figure 4, consistent with the discussion in Section 5.2. We added these new visualizations in Appendix D and Figure 5. Thanks!
>
>
>
> **[Initialization of Prompt Tokens]**
>
> The prompt tokens are sampled from a vocabulary $V$ as described in Section 2 Vocabulary Construction. Initially, they are sampled from a uniform distribution over $V$.
>
> We have also addressed other typos (e.g., Page 7) in the updated manuscript. Thank you for your very constructive suggestions!

---

### Comment · Action_Editors · 2022-12-06
**Update manuscript with right style files**

To authors:

Seems the paper isn't using the right TMLR style file, e.g., with wrong fonts.
Could you update the manuscript accordingly?

---

> ### Author Response · Authors · 2022-12-06
> **Updated with right style files**
>
> Dear Action Editor:
>
> Thanks for your suggestion! We have updated our submission with the right style files. Thanks!

---

### Decision · Action_Editors · 2023-01-28

**Recommendation:** Accept as is

**Comment:**

This paper studies a new setting named black-box prompt learning and proposes a method BDPL for the setting, which achieves good performance on several public datasets.

Reviewers are all positive about the paper and their concerns are well addressed through discussions and revisions. I recommend acceptance.

**Audience:**

Yes

**Claims And Evidence:**

Yes

---

> ### Author Response · Authors · 2023-02-26
> **Camera-Ready Version**
>
> Dear AE and reviewers,
>
> We would like to express our sincere gratitude for all the time and effort you put into reviewing our paper. Your suggestions are invaluable in helping us to improve the quality of our work.
> Following your suggestions, we were able to make some important changes to the manuscript, and we have updated our camera-ready version.
>
> Best regards,
>
> The authors